# Influenza Vaccination and Risk of SARS-CoV-2 Infection in a Cohort of Health Workers

**DOI:** 10.3390/vaccines8040611

**Published:** 2020-10-15

**Authors:** Iván Martínez-Baz, Camino Trobajo-Sanmartín, Irati Arregui, Ana Navascués, Marta Adelantado, Juan Indurain, Ujué Fresán, Carmen Ezpeleta, Jesús Castilla

**Affiliations:** 1Instituto de Salud Pública de Navarra–IdiSNA, 31003 Pamplona, Spain; imartinba@navarra.es (I.M.-B.); ujue.fresan.salvo@navarra.es (U.F.); 2CIBER Epidemiología y Salud Pública (CIBERESP), 28029 Madrid, Spain; 3Clinical Microbiology Department, Complejo Hospitalario de Navarra–IdiSNA, 31008 Pamplona, Spain; camino.trobajo.sanmartin@navarra.es (C.T.-S.); irati.arregui.garcia@navarra.es (I.A.); ana.navascues.ortega@navarra.es (A.N.); marta.adelantado.lacasa@navarra.es (M.A.); juan.indurain.bermejo@navarra.es (J.I.); cezpeleb@navarra.es (C.E.)

**Keywords:** SARS-CoV-2, COVID-19, influenza vaccination, health workers, pandemic

## Abstract

Vaccines may induce positive non-specific immune responses to other pathogens. This study aims to evaluate if influenza vaccination in the 2019–2020 season had any effect on the risk of SARS-CoV-2 confirmed infection in a cohort of health workers. During the first SARS-CoV-2 epidemic wave in Spain, between March and May 2020, a cohort of 11,201 health workers was highly tested by RT-qPCR and/or rapid antibody test when the infection was suspected. Later in June, 8665 of them were tested for total antibodies in serum. A total of 890 (7.9%) health workers were laboratory-confirmed for SARS-CoV-2 infection by any type of test, while no case of influenza was detected. The adjusted odds ratio between 2019–2020 influenza vaccination and SARS-CoV-2 confirmed infection was the same (1.07; 95% CI, 0.92–1.24) in both comparisons of positive testers with all others (cohort design) and with negative testers (test-negative design). Among symptomatic patients tested by RT-qPCR, the comparison of positive cases and negative controls showed an adjusted odds ratio of 0.86 (95% CI, 0.68–1.08). These results suggest that influenza vaccination does not significantly modify the risk of SARS-CoV-2 infection. The development of specific vaccines against SARS-CoV-2 is urgent.

## 1. Introduction

In December 2019, a new coronavirus, named SARS-CoV-2, emerged as the cause of the COVID-19 pandemic [1]. Health institutions and experts are promoting the increase of the influenza vaccine coverage in the 2020–2021 season to reduce the stress on hospitals if the COVID-19 and influenza epidemics overlap and peak around the same time [2,3,4].

Health workers (HWs) are at high risk of acquiring influenza and COVID-19 due to their exposure to patients [5,6]; therefore, HWs are a priority target population for influenza vaccination.

There is no conclusive evidence about the association between influenza vaccination and the risk of SARS-CoV-2 infection. Vaccines may induce positive non-specific immune response to other pathogens [7]. On the other hand, it has been suggested that there is an increased risk of non-influenza respiratory virus infection among influenza vaccine recipients [8]. Several studies have reported that influenza vaccination does not negatively affect the risk of non-influenza respiratory viral infections [9,10,11], but this has not been evaluated for SARS-CoV-2 infection. An ecological study has found high influenza vaccine coverage in COVID-19 cases [12], while others have related influenza vaccination with reduced risk of COVID-19 [13,14,15] and COVID-19 mortality [16,17]. Therefore, studies with more robust design are necessary to evaluate the possible effect of influenza vaccination on COVID-19 outcomes.

Given the lack of specific studies, we aimed to evaluate if influenza vaccination in the 2019–2020 season had any effect on the risk of SARS-CoV-2 confirmed infection in a cohort of HWs.

## 2. Materials and Methods

### 2.1. Study Design and Information Sources

A prospective cohort and a test-negative case-control design were both performed in the present study. On first March 2020, we prospectively defined the cohort of workers of the Navarre Health Service in northern Spain. In October and November 2019, the trivalent inactivated vaccine had been offered free of charge to all HWs.

Since late February 2020, all HWs with COVID-19 symptoms were tested by quantitative real-time reverse transcription polymerase chain reaction (RT-qPCR) in nasopharynx samples or by antibody rapid test (Wondfo, Guangzhou, China) in blood samples. The validity of this rapid test has been evaluated in another study [18]. These tests were also offered by screening protocol to HWs when risk exposure was suspected and in moments of high transmission. This study included result of tests performed from March 1 to May 31, 2020. After the main epidemic period, between June 15 and 30, serological testing was offered to all HWs to detect total antibodies against SARS-CoV-2 by electrochemiluminescence immunoassay (Elecsys^®^, Roche Diagnostics GmbH, Penzberg, Germany).

The circulation of influenza virus in Navarre markedly declined since late February, and the first case of COVID-19 in HWs was confirmed on March 10, the same week that the influenza epidemic returned to baseline incidence. However, RT-qPCR for influenza virus was performed in patients with influenza-like-illness (ILI) who consulted in hospitals or sentinel general practitioners.

Age, sex, major chronic conditions, and ILI diagnoses in the previous five years were obtained from healthcare databases and influenza vaccination status in the 2019–2020 season from the regional vaccination register [19]. The type of profession was considered in the following categories: nursing, nursing assistant, doctor, orderly, and others.

### 2.2. Ethics and Consent Form

Serum samples were obtained with signed consent and swabs with verbal consent of participants. The analysis was done with anonymous data. The study protocol was approved by the Ethical Committee for Clinical Research of Navarre.

### 2.3. Statistical Analysis

The main analysis considered the entire prospective cohort. Influenza vaccination status in the 2019–2020 season was compared between HWs who were confirmed for SARS-CoV-2 by any test and all the others, including those tested negative and those not tested, since the latter had no suspected infection. Sensitivity analyses were done considering only negative testers in the comparison group (test-negative design), considering only one type of test, and including only symptomatic confirmed cases.

Characteristics of HWs by SARS-CoV-2 infection diagnosis were compared using χ^2^ test. Crude and adjusted odds ratios (aOR) and their 95% confidence intervals (CI) were calculated by logistic regression to evaluate the association between influenza vaccination in the 2019–2020 season and the risk of SARS-CoV-2 confirmed infection. Models were adjusted for age groups (18–34, 35–44, 45–54, and ≥55 years), sex, major chronic conditions, profession, and any ILI diagnosis in the previous five years.

## 3. Results

Among 11,201 HWs, 9745 (87.0%) were tested by RT-qPCR (*n* = 8109) and/or antibody rapid test (*n* = 8111), of which 741 (7.6%) were confirmed for SARS-CoV-2 (6.6% of all HWs). COVID-19 symptoms were reported by 3232 (28.9%) HWs, and 503 (15.6%) of them were confirmed for SARS-CoV-2. After the first epidemic wave, serological samples from 8665 (77.4%) HWs were tested, and antibodies against SARS-CoV-2 were detected in 637 (7.4%). A total of 10,555 (94.2%) HWs had a RT-qPCR or rapid test during the epidemic wave or a serological antibody test when the epidemic was over, and 890 (8.4%) of them had any positive test results (7.9% of all HWs). Out of all HWs, only 5.8% were never tested, but this percentage was higher among those aged 55 years or older (6.8%; *p* = 0.007), with major chronic conditions (7.0%, *p* = 0.003), and orderlies (11.4%, *p* < 0.001) (Appendix A).

RT-qPCR or antibody rapid test results showed statistically significant differences by age, type of professional, presence of symptoms, and influenza vaccination status. Serological antibody test results were associated with age, sex, and symptoms, but not with influenza vaccination status (Table 1).

During the study period, 13 HWs who presented ILI symptoms were tested by RT-qPCR for influenza virus and all of them tested negative. Influenza-vaccinated HWs that were confirmed for SARS-CoV-2 by RT-qPCR had a median time from vaccination to COVID-19 diagnosis of 181 days (range, 77–289 days; interquartile range, 161–197 days).

In all multivariate analyses, no statistically significant association was observed between influenza vaccination status and SARS-CoV-2 confirmed infection. In the comparison of HWs with any positive test against the rest of HWs, the aOR for influenza vaccination was 1.07 (95% CI, 0.92–1.24; *p* = 0.387) and for any ILI diagnosis in prior seasons was 0.93 (95% CI, 0.76–1.14; *p* = 0.494).

The comparison of symptomatic patients with a RT-qPCR positive result against those never positive or not tested HWs provided an aOR for influenza vaccination of 1.03 (95% CI, 0.83–1.27; *p* = 0.808). Influenza vaccination also showed no association with SARS-CoV-2 confirmed infection in the comparison of symptomatic patients with a positive result to any test and those never positive or not tested HWs (aOR: 1.09; 95% CI, 0.91–1.31; *p* = 0.334) (Table 2).

Similar results were obtained from the comparison of HWs who had any positive result in RT-qPCR or rapid test against the rest of HWs (aOR: 1.11; 95% CI, 0.94–1.30; *p* = 0.220), and from the comparison of those with a positive result in the serological test against the rest (aOR: 0.99; 95% CI, 0.83–1.18; *p* = 0.930).

Among symptomatic patients tested by RT-qPCR, the classical test-negative design that compared positive cases and negative controls found an aOR of 0.86 (95% CI, 0.68–1.08; *p* = 0.182). Other analyses using only negative testers in the comparison group showed consistent results (Table 2).

## 4. Discussion

In a prospective cohort of HWs, we found that influenza vaccination in the 2019–2020 season did not modify the risk of SARS-CoV-2 confirmed infection during winter and spring 2020. Similar findings were observed in both analyses that considered results of only RT-qPCR or rapid test performed during the epidemic and analyses only based on total antibody test results performed after the epidemic. No association, either, was found in the analyses of the vaccination effect on symptomatic COVID-19 cases in the test-negative design. Influenza vaccination was associated with a higher risk of SARS-CoV-2 infection in some unadjusted analyses, but this association disappeared in adjusted analyses, indicating that confounding factors may explain that spurious association.

Medical consultation for ILI in the previous five seasons was not associated with the risk of SARS-CoV-2 infection, suggesting that previous respiratory viral infections, and mainly influenza infection, do not provide lasting cross-protection against COVID-19. This result does not rule out some short-term cross-protection of one respiratory viral infection against infection with another virus through stimulation of antiviral defenses in the airway mucosa [20].

There is no clear scientific explanation for a possible effect of influenza vaccination on the risk of SARS-CoV-2 infection, but a discussion about potential benefits or risks of influenza vaccination on the risk of COVID-19 is open [12,13,14,15,16,17]. There are immunologic mechanisms by which vaccines against other agents, such as influenza virus, may hypothetically prevent SARS-CoV-2 infection [21]. Influenza haemagglutinin included in influenza vaccines is not related to any of the SARS-CoV-2 antigens, therefore cross-immunity between specific antibodies against both viruses is not expected. However, the cellular immune response against one virus may produce unspecific protection against other infections, but this effect is worse known [21]. Although we did not find evidence for cross-protection, further epidemiological and immunological studies are necessary.

Even in absence of cross-protection, influenza vaccination may be very useful in the COVID-19 pandemic, since by preventing influenza cases that are potentially co-occurring with COVID-19 and leading to worse prognoses, and by reducing the overlap of both epidemics, influenza vaccination can reduce the burden on healthcare systems [20].

As influenza vaccine does not prevent SARS-CoV-2 infection, the development and use of specific vaccines is urgent to control the COVID-19 pandemic.

The strengths of this study are the prospective cohort design in a highly tested population, the laboratory-confirmed diagnosis in all cases, the vaccination status obtained independently of the SARS-CoV-2 results, and the analyses adjusted for the main potential confounding factors. The small incidence of SARS-CoV-2 infection and the high proportion of the study population that was tested may explain the consistent results of the cohort analysis and the test-negative design. In the test-negative design, symptomatic cases were compared with symptomatic controls recruited in similar circumstances before knowing the laboratory result, a fact that reduced selection bias.

This study also has some limitations. Given the study size, we cannot rule out the existence of a possible small effect of influenza vaccination on the risk of SARS-CoV-2 infection. Some HWs were not tested and negative tests do not rule out the infection; however, results of all sensitivity analyses were consistent. Since no deaths occurred in the study population, we cannot conclude about the effect of influenza vaccination on the COVID-19 severity that other authors have suggested [16,17]. As the time between vaccination and SARS-CoV-2 diagnosis was longer than 5 months in most cases, we cannot rule out some short-live effect. Influenza cases should be excluded from the study to prevent bias in the effect estimate [9]. HWs with ILI were tested for influenza and none of them were positive. Furthermore, influenza activity in Navarre was very low during the study period. As only ILI patients were tested for influenza, we cannot totally rule out the inclusion of any influenza case in the study; however the information from epidemiological surveillance by those dates suggests that their number would be low and its impact on estimates would be small.

## 5. Conclusions

These results suggest that influenza vaccination does not significantly modify the risk of SARS-CoV-2 infection. This supports the reinforcement of influenza vaccination programs to prevent influenza infections and their consequences during the COVID-19 pandemic. Furthermore, as influenza vaccine seems not to prevent COVID-19, the development and use of specific vaccines against SARS-CoV-2 is urgent.

## Figures and Tables

**Table 1 vaccines-08-00611-t001:** Characteristics of health workers according to the SARS-CoV-2 infection diagnosis by type of test.

		RT-qPCR or Antibody Rapid Test			Total Antibody Serological Test		
	Total *n* (%) *	Tested *n* (%) **	Positive *n* (%) ***	*p* Value	Tested *n* (%) **	Positive *n* (%) ***	*p* Value
**Total**	11,201 (100)	9745 (87.0)	741 (7.6)		8665 (77.4)	637 (7.4)	
**Age group, years**				<0.001			0.001
18–34	1984 (17.7)	1762 (88.8)	153 (8.7)		1476 (74.4)	133 (9.0)	
35–44	2659 (23.7)	2351 (88.4)	145 (6.2)		2067 (77.7)	121 (5.9)	
45–54	3484 (31.1)	3060 (87.8)	212 (6.9)		2796 (80.3)	190 (6.8)	
≥55	3074 (27.4)	2572 (83.7)	231 (9.0)		2326 (75.7)	193 (8.3)	
**Sex**				0.507			0.003
Male	2083 (18.6)	1805 (86.7)	144 (8.0)		1438 (69.0)	133 (9.2)	
Female	9118 (81.4)	7940 (87.1)	597 (7.5)		7227 (79.3)	504 (7.0)	
**Major chronic conditions**				0.153			0.125
No	8673 (77.4)	7582 (87.4)	561 (7.4)		6753 (77.9)	481 (7.1)	
Yes	2528 (22.6)	2163 (85.6)	180 (8.3)		1912 (75.6)	156 (8.2)	
**ILI diagnosis in the previous 5 years**				0.680			0.181
No	9580 (85.5)	8354 (87.2)	639 (7.6)		7393 (77.2)	555 (7.5)	
Yes	1621 (14.5)	1391 (85.8)	102 (7.3)		1272 (78.5)	82 (6.4)	
**2019–20 season influenza vaccination**				0.006			0.647
No	7348 (65.6)	6422 (87.4)	454 (7.1)		5559 (75.7)	414 (7.4)	
Yes	3853 (34.4)	3323 (86.2)	287 (8.6)		3106 (80.6)	223 (7.2)	
**Type of professional**				<0.001			0.624
Nursing	3573 (31.9)	3181 (89.0)	257 (8.1)		2927 (81.9)	219 (7.5)	
Nursing assistant	1744 (15.6)	1569 (90.0)	103 (6.6)		1345 (77.1)	101 (7.5)	
Doctor	2133 (19.0)	1895 (88.8)	184 (9.7)		1615 (75.7)	128 (7.9)	
Orderly	607 (5.4)	508 (83.7)	30 (5.9)		394 (64.9)	31 (7.9)	
Others	3144 (28.1)	2587 (82.4)	166 (6.4)		2377 (75.8)	157 (6.6)	
**COVID-19 symptoms**				<0.001			<0.001
No	7969 (71.1)	6513 (81.7)	238 (3.7)		5521 (69.3)	230 (4.2)	
Yes	3232 (28.9)	3232 (100)	503 (15.6)		2654 (81.5)	404 (15.3)	
**COVID-19 hospitalization**				<0.001			<0.001
No	11,144 (99.5)	9688 (86.9)	684 (7.1)		8619 (77.3)	592 (6.9)	
Yes	57 (0.5)	57 (100)	57 (100)		46 (80.7)	45 (97.8)	

ILI = influenza-like illness; RT-qPCR = quantitative real-time reverse transcription polymerase chain reaction; rapid test = rapid test for detection of antibodies against SARS-CoV-2 in serum; serological test = detection of total antibodies against SARS-CoV-2 by electrochemiluminescence immunoassay in serum. * Percentage of all health workers. ** Percentage of health workers in each category. *** Proportion or positive individuals among those tested.

**Table 2 vaccines-08-00611-t002:** Association between influenza vaccination in the 2019–2020 season and SARS-CoV-2 infection by type of diagnostic test.

Analysis and Vaccination Status	SARS-CoV-2 Positive/Others	Crude Odds Ratio, (95% CI)	Adjusted Odds Ratio, (95% CI) *	*p* Value
**RT-qPCR, rapid test or serological test (any positive vs. no positive or not tested)**				
Unvaccinated	559/6789	1	1	
Vaccinated	331/3522	1.14 (0.99–1.32)	1.07 (0.92–1.24)	0.387
**RT-qPCR, rapid test or serological test (any positive vs. all negative)**				
Unvaccinated	559/6361	1	1	
Vaccinated	331/3304	1.14 (0.99–1.31)	1.07 (0.92–1.24)	0.363
**RT-qPCR test (positive with symptoms vs. no positive or not tested **)**				
Unvaccinated	248/6789	1	1	
Vaccinated	155/3522	1.21 (0.98–1.48)	1.03 (0.83–1.27)	0.808
**RT-qPCR test in symptomatic patients (positive vs. negative)**				
Unvaccinated	248/1308	1	1	
Vaccinated	155/871	0.94 (0.76–1.17)	0.86 (0.68–1.08)	0.182
**RT-qPCR, rapid test or serological test (any positive with symptoms vs. no positive or not tested)**				
Unvaccinated	331/6789	1	1	
Vaccinated	217/3522	1.26 (1.06–1.51)	1.09 (0.91–1.31)	0.344
**RT-qPCR, rapid test or serological test (any positive with symptoms vs. all negative)**				
Unvaccinated	331/6361	1	1	
Vaccinated	217/3304	1.26 (1.06–1.51)	1.01 (0.91–1.32)	0.334
**RT-qPCR or rapid test (any positive vs. no positive or not tested)**				
Unvaccinated	454/6894	1	1	
Vaccinated	287/3566	1.22 (1.05–1.43)	1.11 (0.94–1.30)	0.220
**RT-qPCR or rapid test (any positive vs. all negative)**				
Unvaccinated	454/5968	1	1	
Vaccinated	287/3036	1.24 (1.07–1.45)	1.13 (0.96–1.32)	0.141
**Total antibody serological test (positive vs. no positive or not tested)**				
Unvaccinated	414/6934	1	1	
Vaccinated	223/3630	1.03 (0.87–1.22)	0.99 (0.83–1.18)	0.930
**Total antibody serological test (positive vs. all negative)**				
Unvaccinated	414/5145	1	1	
Vaccinated	223/2883	0.96 (0.81–1.14)	0.93 (0.78–1.11)	0.413

CI = confidence interval; RT-qPCR = quantitative reverse transcription polymerase chain reaction; rapid test = rapid test for detection of antibodies against SARS-CoV-2 in serum; serological test = detection of total antibodies against SARS-CoV-2 by electrochemiluminescence immunoassay in serum. * Logistic regression model adjusted for age groups (18–34, 35–44, 45–54, and ≥55 years), sex, major chronic condition, any influenza-like illness diagnosis in the previous five years, and type of professional. ** No positive test results for any of the tests performed: RT-qPCR, rapid test, or serological test.

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
