# Peer review of "Influenza Vaccination and Risk of SARS-CoV-2 Infection in a Cohort of Health Workers"

_vaccines, 2020, doi:10.3390/vaccines8040611_

Round 1
Reviewer 1 Report
This study is relevant and comes at a rime when other publications have discussed the same issue.
The study is based on a quality data set of the Navarra region (Spain) that is already known in the field of influenza and has been used for various study designs (cohort , case control and TND studies) for vaccine effectiveness studies.
The main message from the study is that there is no significant association between 2019-20 influenza seasonal vaccination and occurrence of SARS-Cov 2 infection. This is supported by the various study designs and analyses used by the authors which lead to similar results.
Suggestions
Authors may want to describe the characteristics of persons who were not tested, or of those who were not included in the study.
Authors could analyse the effect of influenza vaccination only against symptomatric PCR + SARS-COV2 (excluding cases that are only positive to rapid tests or serology).
I understand that most of the HW cohort was tested. This means that, due to the small incidence of SARS Cov 2 infection, those testing negative may be representative of the influenza vaccine coverage in the entire cohort. This may be why cohort and TND analyses give similar results.
However, it is not clear if authors have attempted to verify if keeping the influenza cases in the control group violate one of the assumptions of TND studies. Specificaly the risk of "non SARS -Cov 2 ARI or ILI" should be equal between those vaccinated against influenza and not. It is important to verify if keeping infuenza cases in the TND control group, the likelyhood of controls to be vaccinated decreases and if this control group is still representative of the influenza vaccine coverage in the source population of SARS-Cov-2 cases.
However unlikely if everyone is tested, authors have an opportunity to verify this assumption. Using their entire cohort, authors could verify that the risk of Non SARS Cove 2 ARI/ILI is the same among those vaccinated and unvaccinated against influenza. They could also complete their TND analysis by re-doinfg the analysis excluding from the control group influenza cases that occured earlier in the season.
With those additional analyses this article could become a good contribution to the current debate.
It would also be nice showing if anyone in the data base had had a double infection in the season (SARS Cov 2 and influenza).
Author Response
RESPONSE TO REVIEWER #1
Authors may want to describe the characteristics of persons who were not tested, or of those who were not included in the study.
AUTHORS’ RESPONSE: The proportion of HWs who were not tested is presented in results (line 106-109) “Out of all HWs only 5.8% were never tested, but this percentage was higher among those aged 55 years or older (6.8%; p=0.007), with major chronic conditions (7.0%, p=0.003), and orderly (11.4%, p<0.001) (Supplementary Tables S1 and S2)”. Additionally, we have added supplementary tables S1 and S2.
Authors could analyse the effect of influenza vaccination only against symptomatric PCR + SARS-COV2 (excluding cases that are only positive to rapid tests or serology).
AUTHORS’ RESPONSE: We have added this analysis in table 2 and in results (lines 184-185) we say: “Among symptomatic patients tested by RT-qPCR, the original test-negative design that compared positive cases and negative controls found an aOR of 0.86 (95% CI, 0.68-1.08; p=0.182)”.
I understand that most of the HW cohort was tested. This means that, due to the small incidence of SARS Cov 2 infection, those testing negative may be representative of the influenza vaccine coverage in the entire cohort. This may be why cohort and TND analyses give similar results.
AUTHORS’ RESPONSE: Now, we explain this point in the discussion “The small incidence of SARS-CoV-2 infection and the high proportion of the study population that was tested may explain the consistent results of the cohort analysis and the test-negative design.” (lines 261-263)
However, it is not clear if authors have attempted to verify if keeping the influenza cases in the control group violate one of the assumptions of TND studies. Specificaly the risk of "non SARS -Cov 2 ARI or ILI" should be equal between those vaccinated against influenza and not. It is important to verify if keeping infuenza cases in the TND control group, the likelyhood of controls to be vaccinated decreases and if this control group is still representative of the influenza vaccine coverage in the source population of SARS-Cov-2 cases.
AUTHORS’ RESPONSE: In material and method section we have added this sentence “The circulation of influenza virus in Navarre markedly declined since late February and the first case of COVID-19 in HWs was confirmed on March 10, the same week that the influenza epidemic returned to baseline incidence. However, RT-qPCR for influenza virus was performed in patients with influenza-like-illness (ILI) who consulted in hospitals or sentinel general practitioners.” (lines 66-69). In results we have added: “During the study period, 13 HWs who presented ILI symptoms were tested by RT-qPCR for influenza virus and all of them tested negative.” (lines 163-164). We also recognize this limitation “Influenza cases should be excluded from the study to prevent bias in the effect estimate [9]. HWs with ILI were tested for influenza and none of them was positive. Furthermore, influenza activity in Navarre was very low during the study period. As only ILI patients were tested for influenza, we cannot totally rule out the inclusion of any influenza case in the study; however the information from epidemiological surveillance by those dates suggests that their number would be low and its impact on estimates would be small.” (lines 277-282).
In the abstract, we have added, “…while no case of influenza was detected”
However unlikely if everyone is tested, authors have an opportunity to verify this assumption. Using their entire cohort, authors could verify that the risk of Non SARS Cove 2 ARI/ILI is the same among those vaccinated and unvaccinated against influenza. They could also complete their TND analysis by re-doinfg the analysis excluding from the control group influenza cases that occured earlier in the season.
With those additional analyses this article could become a good contribution to the current debate.
It would also be nice showing if anyone in the data base had had a double infection in the season (SARS Cov 2 and influenza).
AUTHORS’ RESPONSE:
We agree your point. Using the entire cohort, we have compared the risk of non SARS-CoV-2 ARI/ILI among those vaccinated and unvaccinated against influenza, and we have found an aOR= 1.16 (95% CI, 1.06-1.28). This statistically significant result may be explained because influenza vaccine coverage is higher among people prone to ARI/ILI. This association supposes a residual confounding difficult to be controlled. However, since it would only affect some of our analysis but not to others, and all the different analysis that we have done have consistent results, we would prefer do not open this methodological discussion in our manuscript.
Regarding the second point, we say in the abstract: “…while no case of influenza was detected.”

Reviewer 2 Report
The authors arrive to the conclusion that influenza vaccination does not significantly modify (positively or negatively) the risk of SARS-CoV-2 infection in the population of the study. This is relevant to support the reinforcement of influenza vaccination during the COVID-19 pandemic although (as mentioned by the authors) there is no conclusion of the effect of influenza vaccination on the COVID-19 severity, as shown by other authors.
I will comment about few points:
1- You state that "..influenza vaccination was associated with a higher risk of SARS-CoV-2 infection in some unadjusted analyses, but this association disappeared in adjusted analyses, indicating that confounding factors may explain that association". Due to the importance of this, could it be possible to discuss further and/or provide more (specific) elements on the adjustments conducted and how the data were treated to move from the crude odds ratio to the adjusted odds ratio as shown in Table 2??
2- I felt that the abstract could have been more explicative as the first contact of the reader with the study. It has 165 words and there is still room for more information which can be found in the body of the manuscript.
3- Is there data available/published on the correlation of results from real-time RT-qPCR in nasopharynx samples and antibody rapid tests (from Wondfo, Guangzhou, China) in blood samples?.
Author Response
RESPONSE TO REVIEWER #2
The authors arrive to the conclusion that influenza vaccination does not significantly modify (positively or negatively) the risk of SARS-CoV-2 infection in the population of the study. This is relevant to support the reinforcement of influenza vaccination during the COVID-19 pandemic although (as mentioned by the authors) there is no conclusion of the effect of influenza vaccination on the COVID-19 severity, as shown by other authors.
I will comment about few points:
1- You state that "..influenza vaccination was associated with a higher risk of SARS-CoV-2 infection in some unadjusted analyses, but this association disappeared in adjusted analyses, indicating that confounding factors may explain that association". Due to the importance of this, could it be possible to discuss further and/or provide more (specific) elements on the adjustments conducted and how the data were treated to move from the crude odds ratio to the adjusted odds ratio as shown in Table 2??
AUTHORS’ RESPONSE: Analyses were adjusted for all potential confounding factors that were available. In lines 94-95 we say, “Models were adjusted by age groups (18–34, 35–44, 45–54 and ≥55 years), sex, major chronic conditions, profession, and any ILI diagnosis in previous five years.” In results section, we say, “RT-qPCR or antibody rapid test results showed statistically significant differences by age, type of professional, presence of symptoms, and influenza vaccination status. Serological antibody test results were associated with age, sex, and symptoms, but not with influenza vaccination status (Table 1).” In several analyses, both crude and adjusted estimates were not statistically significant and differences between both estimates were small in all cases.
2- I felt that the abstract could have been more explicative as the first contact of the reader with the study. It has 165 words and there is still room for more information which can be found in the body of the manuscript.
AUTHORS’ RESPONSE: New sentences have been added: “while no case of influenza was detected”, “Among symptomatic patients tested by RT-qPCR, the comparison of positive cases and negative controls showed an adjusted odds ratio of 0.86 (95% CI, 0.68-1.08).” and “The development of specific vaccines against SARS-CoV-2 is urgent.”
3- Is there data available/published on the correlation of results from real-time RT-qPCR in nasopharynx samples and antibody rapid tests (from Wondfo, Guangzhou, China) in blood samples?.
AUTHORS’ RESPONSE: Since RT-qPCR assess the virus presence and antibody rapid test detect antibodies, they detect different moments in the infection. The low correlation does not mean that rapid test had not good sentitivity and especificity. We say in methods “The validity of this rapid test has been evaluated in another study [18]” (lines 60-61), and we provide a new reference number 18 that compares antibody rapid test and total antibody serological test.